# The power of Dionysus—Effects of red wine on consciousness in a naturalistic setting

**Rui Miguel Costa** [1]*, **Arlindo Madeira**[2,3], **Matilde Barata** [4], **Marc Wittmann** [5]

**1** William James Center for Research, ISPA – Instituto Universitário, Lisbon, Portugal, **2** Tourism and Hospitality Management School, Universidade Europeia, Lisbon, Portugal, **3** ESCAD—Escola Superior de Ciências e Administração, Lisbon, Portugal, **4** ISPA – Instituto Universitário, Lisbon, Portugal, **5** Institute for Frontier Areas of Psychology and Mental Health, Freiburg, Germany

* rcosta@ispa.pt

## Abstract

There is lack of research on effects of red wine on consciousness when drank in wine bars designed to enhance the pleasurableness of the wine drinking experience. Effects of a moderate dose of red wine ($\approx$ 40.98 g of ethanol) on consciousness were examined in a naturalistic study taking place in a wine bar located in one of the most touristic areas of Lisbon, Portugal. One hundred two participants drank in one of three conditions: alone, in dyad, or in groups up to six people. Red wine increased pleasure and arousal, decreased the awareness of time, slowed the subjective passage of time, increased the attentional focus on the present moment, decreased body awareness, slowed thought speed, turned imagination more vivid, and made the environment become more fascinating. Red wine increased insightfulness and originality of thoughts, increased sensations of oneness with the environment, spiritual feelings, all-encompassing love, and profound peace. All changes in consciousness occurred regardless of volunteers drinking alone, in dyad or in group. Men and women did not report different changes in consciousness. Older age correlated with greater increases in pleasure. Younger age correlated with greater increases in fascination with the environment of the wine bar. Drinking wine in a contemporaneous Western environment designed to enhance the pleasurableness of the wine drinking experience may trigger changes in consciousness commonly associated with mystical-type states.

*Bronze is the mirror of the form; wine, of the heart*

*Aeschylus*

*Let's gallop on the steeds of wine to heavens magic and divine*

*Baudelaire*

## Introduction

Altered states of consciousness refer to substantial deviations from the habitual waking consciousness. Among the common human needs, there is search for pleasant altered states of

**Data Availability Statement:** The dataset is now publicly available at https://osf.io/a72ep/.

**Funding:** The authors received no specific funding for this work.

**Competing interests:** The authors have declared that no competing interests exist.

consciousness, that is, a temporary joyful transcendence of the ordinary mental state [1,2]. The balanced consumption of wine can be a means to such joys, one that is deeply ingrained in many human cultures since time immemorial [3].

Red wine is an alcoholic beverage made from the fermentation of dark grapes whose alcohol by volume commonly varies between 12% and 15%. Among the many alcoholic beverages, red wine is one of the oldest, and it is the most connected to the appreciation of meals, and the most connected to hedonism (e.g., restaurants specialized in red wine tend to offer a more hedonic environment; red wine is a drink of choice in romantic dinners) [4–7]. Once regarded as a luxury good, wine has been democratized and is enjoyed by a much wider socio-economic range of increasingly sophisticated consumers [8]. Nowadays, it is also appreciated by a greater variety of ethnical groups [9–11], and drinking it with meals was associated with better psychological well-being [9]. The hedonistic nature of red wine involves the indulgence of the senses in wine products and in the esthetic framework of the wine landscape, which encompasses wine bars [12]. In addition, red wine is the most studied alcoholic beverage regarding the cognitive and perceptual factors that affect how it tastes [13,14]. The extensive research on factors influencing taste contrasts markedly with the dearth of research on pleasant altered states of consciousness induced by moderate doses of red wine, especially from a first-person introspective perspective [15]. The esthetics of places where wine is drunk affect the experience of wine appreciation [12,16–18]. Therefore, naturalistic designs that reflect typical drinking experiences are important when studying altered states of consciousness elicited by red wine.

With implemented control conditions and standardized procedures, randomized controlled trials are useful to study effects of particular interventions. However, sometimes they are more limited with regards to ecological validity. This applies to the drinking experience in wine bars to where very few people will willingly go to drink a non-alcoholic beverage, and whose esthetics and environment are not thought to engage drinkers of non-alcoholic beverages, making control conditions unfeasible or highly artificial. Although naturalistic studies lack strict standardization and control conditions, they allow an exploration of the effects of wine in the real-world circumstances where wine is usually consumed; this increases ecological validity. There is research indicating that drinking wine in wine bars generates an experience subjectively different from the one occurring when drinking in laboratories or otherwise non-naturalistic settings [16]. Surprisingly, there is a lack of studies on how a moderate amount of red wine influences consciousness when the wine is drank in wine bars, which are designed to enhance the pleasurableness of the red wine drinking experience. In order to better understand the effects of red wine, it is of essence to conduct research in the context of the wine bar experience, which allows visitors not just to taste quality wine, but also to get immersed into an environment where it is possible to relax and socialize during tasting [12,16].

Core features of altered states of consciousness are changes in the awareness of body, space, and time [1,2,19,20]. Altered states of consciousness are often characterized by creative outbursts, more vivid mental images, and enhancement of the perceived beauty of external environment [21,22]. Consistently with this view, alcohol is frequently seen as a promoter of creativity and inspiration. This is confirmed by laboratory studies showing that moderate doses of beer or vodka increase creative problem solving [23,24], although alcohol-related creativity was also found to be enhanced by expectations rather than actual effects of alcohol [25]. Alcohol also appears to enhance the beauty of the environment, as a laboratory study found that moderate consumption of vodka increased the attractiveness of faces and landscapes [26]. Nevertheless, there is lack of research with naturalistic designs about the effects of red wine on insightfulness and changes in the perception of the environment.

Altered states of consciousness are sometimes characterized by mystical experiences, that is, a sense of connection with another realm of reality outside of common space-time that is

difficult to translate into words. These inner states are characterized by feelings of eternity, connection with an invisible higher power, perceiving things more real than real, and dissolution of borders between self and world [27–32]. The sense of lack of separation between self and world is often given the name oceanic feeling, a term known to be first used by the French writer and mystic Romain Rolland in a letter to Sigmund Freud, who later described it as ". . . a sensation of 'eternity', a feeling as if of something limitless, unbounded—as it were, 'oceanic' . . . a feeling of an indissoluble bond, of being one with the external world as a whole" (p. 64) [33]. Freud discussed oceanic feelings as possible origins of spiritual feelings that can find existence outside institutional religions [33]. Later authors identified oceanic feelings as a common feature of mystical experiences, but noting they might be necessary but not sufficient to trigger a complete mystical state [27].

In poetry, wine has been portrayed as a doorway to the divine [34,35], and in ancient mystical traditions, wine was used to promote contact with divinities [36]. The possible effects of alcohol on mystical feelings were noted by William James, often acknowledged as the "father of American psychology", who stated that the "sway of alcohol over mankind is unquestionably due to its power to stimulate the mystical faculties of human nature" (p. 376) [37]. However, there is a notable lack of empirical research on effects of red wine on mystical experiences and oceanic feelings. To the best of our knowledge, only one experimental study examined the effects of alcohol on experiences related to mysticism, and failed to find a relationship [38]. Yet, the authors noted an important limitation: during the experiment, the participants of the study were in conditions of visual and auditory deprivation, and this might have inhibited the triggering of mystical feelings. In fact, alcoholic drinks and red wine in particular are commonly appreciated in environments rich in sensory stimulation, not in sensory deprived environments. This further makes the need of naturalistic studies important.

The present study uses a naturalistic pre-post design with the objective of examining how a moderate dose of red wine induces altered states of consciousness in a group of clients of wine bars comprising mostly university students and professionally active people in their twenties, thirties, and forties. Additionally, it aims at exploring whether the changes in consciousness caused by a moderate dose of red wine differ between three conditions: drinking alone, in groups of two (dyads), and in groups between three and six persons.

## Materials and methods

### Participants and procedure

The study was conducted in a wine bar specialized in selling red wine paired with food in Lisbon. The sample of the study ($N = 102$) was recruited among costumers visiting the bar, frequenters and tourists of the Lisbon touristic area where the bar is located, as well as students and staff of a nearby university, who often came to the experiment with acquaintances who also participated. All potential participants were approached personally by members of the research team and invited for a study investigating how a moderate amount of red wine causes changes in consciousness. They were informed that two glasses of red wine would be offered in the wine bar, and that they would just have to enjoy the moment while drinking. They were additionally informed that they would be asked to complete the questionnaire in English about changes in consciousness. Upon agreement, the participation date was scheduled. There were no other incentives for the participation in the study. The inclusion criteria were being an adult for whom drinking two glasses of red wine is a familiar experience, not having consumed alcohol before, not having consumed psychoactive drugs before, and understanding English. Thirteen participants reported that they usually do not drink red wine (see Table 1), but before

**Table 1. Demographics.**

|  | Percent |
|---|---|
| *Sex* | |
| Women | 55.9 |
| Men | 44.1 |
| *Occupation* | |
| Student | 21.6 |
| Employed | 69.6 |
| Unemployed | 2.0 |
| Retired | 2.0 |
| Did not respond | 4.9 |
| *Usual consumption of red wine* | |
| Usually don't drink | 12.7 |
| Less than 3 glasses a week | 37.3 |
| Between 3 and 10 glasses a week | 42.2 |
| Between 11 and 20 glasses a week | 5.9 |
| More than 21 glasses a week | 2.0 |
| *Usual alcohol consumption* | |
| Usually don't drink | 1.0 |
| Less than 3 glasses a week | 22.5 |
| Between 3 and 10 glasses a week | 51.0 |
| Between 11 and 20 glasses a week | 21.6 |
| More than 21 glasses a week | 3.9 |
| *Usual tobacco consumption* | |
| Non-smokers | 50.0 |
| Less than 10 cigarettes a day | 29.4 |
| More than 10 cigarettes a day | 20.6 |
| *Nationality* | |
| American | 10.8 |
| Brazilian | 2 |
| Bulgarian | 2 |
| Canadian | 2.9 |
| Dutch | 2.9 |
| English | 2 |
| French | 2 |
| French-Swiss | 1 |
| German | 1 |
| Hungarian | 2 |
| Italian | 7.8 |
| Italian-Brazilian | 1 |
| Norwegian | 1 |
| Portuguese | 57.8 |
| South-African | 1 |
| Spanish | 1 |
| Turkish | 2 |

the experiment we confirmed that for them red wine was a familiar experience, because it is occasionally drank.

 The questionnaire was only available in English, as the wine bar and the surrounding area are frequented by many international tourists. Mean age was 35.39 years ($SD = 11.91$);

median = 33; range: 20–70. Descriptive statistics are displayed in Table 1, including the self-reported frequency of alcohol and red wine consumption, which are considered reliable measures [39,40]. The final sample size was obtained after one participant having been discarded, because of difficulties understanding the questionnaire in English.

Participants were asked to drink two glasses (18.5 cl. each) of *Quinta da Lapa Reserva Syrah* 2018, a silky full-bodied red wine from the Lisbon region with 14˚ of alcohol ($\approx$ 20.49 g of ethanol each). The choice of this particular wine was made by the producer and by the two sommeliers of the bar, where the experiment was conducted, based on their know-how of the market. It is a consensual wine with a new world profile and origin in a hot terroir, where grapes have greater maturation. This allows silky, full-bodied wines with a lesser degree of acidity, which are enjoyed by the majority of people. During the two weeks preceding the study, several clients of the bar were offered this wine (for free) for a blind tasting, and the feedback was positive and unanimous relative to its quality.

Wine was served in the same type of glasses and at a temperature of 18 Celsius degrees. Participants did not have to pay for the wine. The participants were instructed that they could drink during the time span they wish. They were also allowed to drink water, eat some snacks and smoke, as the design is naturalistic, and eating and smoking are part of the normal experience of drinking red wine for many people.

In the wine bar, the background music consisted only of classic jazz music. The bar has an intimate atmosphere with clients talking in low voice. There is a maximum occupation of 28 people shared by six tables, each one allowing between four and six people. There are no standing clients.

Participants were asked not to consume other drinks (with exception of water) due to the interference on the flavor. They were also asked not to use smartphones or other technologies due to the interference these could cause on the attention and enjoyment of the experience at the wine bar [41–43], as well as on time perception [44], which is a variable of interest. Notepads and pens were available for those participants who wanted to write. The eaten food consisted entirely of light snacks (bread with cheeses, olive oil, or smoked sausages). All participants provided a written informed consent. The study was approved by the local Ethics Committee.

All participants were sitting at a table during the experiment. Once arrived at the table, they completed a questionnaire on demographics, drinking habits, smoking habits, and several aspects of baseline consciousness, that is, the aspects of the awareness that respondents had of themselves and their surroundings immediately before drinking. Immediately after having finished the second glass, a questionnaire about the same aspects of consciousness was provided, but this time referring to the period in which the wine was drank (see subsection "measures of consciousness before and after the wine"). Participants were allocated to one of three conditions: 1) drinking alone (in solitude), 2) drinking in dyad (couple, same-sex friends, or opposite-sex friends), and 3) drinking in a group, i.e. groups of friends (between three and six persons) (for group statistics, see Table 2). The allocation depended entirely on how participants appeared in the wine bar, that is, alone, in dyad or in a group. In order to maintain the naturalistic design, all participants drinking in dyads or in groups drank with their friends or partners.

The study received the approval of the Ethics Committee of ISPA—Instituto Universitário.

## Measures of consciousness before and after the wine

Pleasure and arousal were assessed by the two respective subscales of the Self-Assessment Manikin (SAM) [45]. Response options varied between 1 (least possible pleasure or arousal) and 9 (greatest possible pleasure or arousal).

**Table 2. Context characteristics.**

|  | % |
| --- | --- |
| *Context of participation* |  |
| In solitude | 30.4 |
| In dyad | 38.2 |
| In group | 31.4 |
| *Smoked during the experiment* |  |
| No | 60.8 |
| Yes | 39.2 |
| *Ate during the experiment* |  |
| No | 17.6 |
| Yes | 82.4 |

Intensity of the awareness of the body before and after drinking was assessed by a visual analogue scale with a picture representing a human figure drawn with lines of different thickness. Responses relative to the figure vary between 1 (the least thick lines) and 7 (the thickest lines [2,28,46–48].

Intensity of awareness of time and the speed of time passage were measured with visual analogue scales in which respondents make a stroke on 100-mm horizontal lines anchored from "Not at all" to "Extremely", and from "Extremely slowly" to "Extremely fast", respectively [2,46].

Perception of the speed of thoughts was measured with a visual analogue scale in which respondents make a stroke on 100-mm horizontal lines anchored from "Extremely slowly" to "Extremely fast".

The differential foci on past, present and future were assessed by asking respondents to make two strokes on a 100-mm horizontal line, in which the space between the left extreme of the line and the first stoke represents the orientation towards memories (past), the space between the first and the second stroke represents the orientation to momentary experience (present) and the space between second stroke and the right extreme of the line refers to the orientation towards expectations and plans (future) [47].

The Altered States of Consciousness Rating Scale (OAV) was used to measure other aspects of changes in consciousness [22]. For the present purpose, we used six dimensions: 1) Experience of Unity (e. g. "Everything seemed to unify into an oneness"), 2) Spiritual Experience (e.g., "I had the feeling of being connected to a superior power"), 3) Blissful State (e. g., "I experienced a profound peace in myself"), 4) Insightfulness (e.g., "I gained clarity into connections that puzzled me before"), 5) Complex Imagery (e.g., "My imagination was extremely vivid", 6) Changed Meaning of Percepts, that is, enhanced fascination with the environment (e.g., "Objects around me engaged me emotionally much more than usual"). The OAV scales were rated on a visual analogue scale with a stroke on a 100-mm horizontal line anchored from "No, no more than usual situations" to "Yes, much more than usual situations". For the purpose of the present study we excluded other dimensions of the Altered States of Consciousness Rating Scale (OAV): Disembodiment, Anxiety, Impaired Control and Cognition, Elementary Imagery, Audio-Visual Synesthesias. The reason is that we did not want to burden participants with questions about unlikely changes such as panic or disembodiment experiences, and because we did not expect that many participants sitting in the wine bar, often socializing, would be attentive to alterations in mental imagery by staying a long time with closed eyes or otherwise in a mental state very detached from the social environment.

## Statistical analyses

Repeated measures ANOVAs were performed for comparing the different aspects of consciousness before and after drinking the wine. Condition of participation (solitude, dyad, group) was entered as between-subjects factor [49]. All analyses were done with SPSS-25.

## Results

As depicted in Table 3, red wine (with 40.98 g of ethanol) improved mood and increased arousal. Red wine diminished the awareness of time, and increased reports of time passing slower. It increased the awareness of the present moment while decreasing the awareness of past memories and expectations about the future. It made thoughts pass slower and decreased body awareness.

Red wine caused significant changes in the six dimensions of the Altered States of Consciousness Rating Scale (OAV): Experience of Unity, Spiritual Experience, Blissful State, Insightfulness, Complex Imagery, and Change in Meaning of Percepts were all increased (see Table 3).

There were no significant interactions with the condition of participation, which shows that the detected effects of red wine occurred in people drinking in solitude, in dyad, and in small groups (see Table 3).

With the aim of examining if the effects of wine differed between men and women, and between Portuguese and foreigners, repeated measures ANOVAs with sex or nationality (Portuguese vs. foreigner) as fixed factors were performed. Lack of significant interactions with sex showed that men and women do not differ in alterations of consciousness (all $p > .05$). There were two significant interactions with nationality: compared to Portuguese, foreigners became more focused on present moment: foreigners' average difference from baseline = 19.12, $SD$ = 24.19 vs. Portuguese average difference from baseline = 9.12, $SD$ = 9.10, $t$ (df) = 2.04 (97), partial $\eta^2$ = .041, $p$ = .044. Compared to Portuguese, foreigners became less focused on plans and expectations about the future: foreigners' average difference from baseline = -13.81, $SD$ = 21.44 vs. Portuguese average difference from baseline = -4.02, $SD$ = 17.46, $t$ (df) = 2.50 (97), partial $\eta^2$ = .061, $p$ = .014. There were no other significant interactions with nationality (all $p > .05$).

In Pearson's correlations, age correlated directly with increases in pleasure ($r$ = .20, $p$ = .045), that is, the older the participants, the greater the tendency to report greater increases of pleasure during the experience of drinking the wine. Age correlated inversely with Change in Meaning of Percepts ($r$ = -.20, $p$ = .042), that is, the younger the participants, the greater the tendency to report that the environment became more fascinating while drinking the wine.

## Discussion

In the present naturalistic study, it was possible to identify meaningful changes in consciousness caused by a moderate dose of red wine. These changes occurred regardless of drinking in solitude, in dyad, or in group. Not surprisingly, red wine improved mood. The improvement in mood was further shown in differences in the OAV subscale Blissful State which has items referring to experiencing profound peace, all-embracing love, and boundless pleasure.

Red wine increased arousal. Ethanol is commonly seen as a sedative substance largely because of its action on the GABAergic system, but ethanol also causes dopamine release in the brain [50], which makes it act to some extent as a stimulant, especially at moderate doses. Additionally, the ethanol present in red wine can increase heart rate [50,51] and muscle sympathetic nerve activity [52], which may contribute to the subjective feeling of arousal. The increases in heart rate correlated with increases in central dopaminergic activity [50]. Such

**Table 3. Effects of red wine on consciousness.**

| | Solitude | Dyad | Group | Total sample | $F$ (df) Partial $\eta^2$ Main effect 'before vs. after wine'[1] | $F$ (df) Partial $\eta^2$ Interaction with 'condition of participation'[1] |
|---|---|---|---|---|---|---|
| Pleasure | 6.42 (1.65) | 6.26 (1.77) | 7.25 (1.50) | 6.62 (1.70) | 59.75 (1, 99) | .97 (2, 99) |
| | 7.48 (1.26) | 7.74 (1.23) | 8.28 (1.05) | 7.83 (1.22) | .38*** | .02 (NS) |
| Arousal | 4.39 (1.52) | 5.26 (1.78) | 5.19 (2.04) | 4.97 (1.82) | 26.45 (1, 97) | .30 (2, 97) |
| | 5.68 (1.68) | 6.16 (2.03) | 6.32 (2.29) | 6.06 (2.01) | .21*** | .01 (NS) |
| Body awareness | 5.70 (.99) | 6.10 (1.07) | 6.22 (.71) | 6.02 (.96) | 8.40 (1, 98) | .29 (2, 98) |
| | 5.47(1.57) | 5.64 (.96) | 5.81 (1.15) | 5.64 (1.22) | .08** | .01 (NS) |
| Time awareness | 48.16 (23.76) | 57.92 (26.27) | 57.88 (22.83) | 54.94 (24.65) | 22.61 (1, 99) | .82 (2, 99) |
| | 36.00 (28.60) | 34.79 (34.34) | 33.47 (30.33) | 34.75 (31.14) | .19*** | .02 (NS) |
| Time speed | 54.13 (21.50) | 59.33 (22.03) | 61.16 (21.97) | 58.34 (21.82) | 15.39 (1, 97) | .01 (2, 97) |
| | 40.77 (27.93) | 45.08 (31.29) | 47.13 (32.63) | 44.42 (30.55) | .14*** | .00 (NS) |
| Past | 27.07 (17.71) | 25.31 (14.92) | 24.27 (13.19) | 25.53 (15.23) | 10.36 (1, 96) | .003 (2, 96) |
| | 22.07 (16.90) | 20.00 (16.40) | 19.13 (14.99) | 20.36 (16.02) | .10*** | .00 (NS) |
| Present | 46.23 (23.05) | 40.41 (17.62) | 44.20 (16.55) | 43.32 (19.10) | 27.64 (1, 96) | .79 (2, 96) |
| | 55.03 (28.88) | 56.59 (22.76) | 58.43 (21.26) | 56.68 (24.15) | .22*** | .02 (NS) |
| Future | 26.70 (14.02) | 34.28 (17.42) | 31.53(11.85) | 31.15 (15.08) | 15.80 (1, 96) | 1.14 (2, 96) |
| | 22.90 (18.38) | 23.41 (14.92) | 22.43(12.66) | 22.96 (15.29) | .14*** | .02 (NS) |
| Thought speed | 55.35 (18.58) | 58.58 (23.57) | 67.35(20.76) | 60.30(21.61) | 7.55 (1, 97) | 2.05 (2, 97) |
| | 53.32 (25.71) | 51.76 (28.35) | 48.87(26.96) | 51.35 (26.91) | .07** | .04 (NS) |
| Experience of Unity | 18.00 (17.47) | 12.12 (14.10) | 23.92 (20.74) | 17.61 (17.94) | 79.04 (1, 99) | 1.12 (2, 99) |
| | 33.78 (24.65) | 35.90(24.34) | 46.44(27.02) | 38.56 (25.63) | .44*** | .02 (NS) |
| Spiritual Experience | 15.67(20.65) | 4.13(5.62) | 17.01 (19.96) | 11.62 (17.19) | 29.22 (1, 98) | 1.14 (2, 98) |
| | 22.99 (22.46) | 13.62 (17.25) | 31.54 (28.73) | 22.00 (23.83) | .23*** | .02 (NS) |
| Blissful State | 19.63 (20.67) | 12.09 (14.59) | 27.27 (22.04) | 19.05 (19.86) | 67.02 (1, 96) | 1.22 (2, 96) |
| | 35.66 (23.72) | 37.82 (26.62) | 49.84 (28.00) | 40.78 (26.62) | .41*** | .03 (NS) |
| Insightfulness | 21.72 (23.23) | 10.64 (15.82) | 20.62 (19.03) | 17.11 (19.81) | 66.67 (1, 98) | .35 (2, 98) |
| | 38.46 (26.59) | 32.09 (21.99) | 39.88 (28.08) | 36.44 (25.63) | .41*** | .01 (NS) |
| Changed | 16.49 (21.92) | 7.74 (11.17) | 19.83 (21.55) | 14.20 (18.98) | 52.47 (1, 99) | .93 (2, 99) |
| Meaning of Percepts | 34.91 (27.04) | 26.79 (24.23) | 31.95 (29.73) | 30.88 (26.85) | .35*** | .02 (NS) |

*(Continued)*

**Table 3.** (Continued)

| | Solitude | Dyad | Group | Total sample | F (df)<br>Partial η²<br>Main effect 'before vs. after wine'[1] | F (df)<br>Partial η²<br>Interaction with 'condition of participation'[1] |
|---|---|---|---|---|---|---|
| Complex Imagery | 13.27 (18.45) | 6.64 (10.06) | 15.67 (18.27) | 11.45 (16.01) | 54.31 (1, 98) | .11 (2, 98) |
| | 25.91 (23.06) | 20.94 (18.74) | 30.44 (25.19) | 25.38 (22.34) | .36*** | .002 (NS) |

[1] The three conditions of participation were a) drinking in solitude, b) drinking in dyad, c) drinking in groups of three to six people.

*** p < .001;

** p < .01;

NS = nonsignificant.

Line above (before wine), line below (after wine).

activation of the dopaminergic neurons is consistent with the documented effects of a moderate dose of alcohol on the reward system by which incentive salience is increased [53,54].

Red wine had substantial effects on time awareness. The awareness of time diminished and time was felt as passing more slowly. It is often noted that good moments pass quickly. In extremely altered states of consciousness such as under the influence of psychedelics and in deep meditative states of experienced practitioners, a loss of sense of self and time are often reported [19,20,32,55]. That does not seem to be the case in the red wine experience: here, pleasantness and some degree of "time dilation" do concur. This most likely has to do with the retrospective judgment of the previous interval of a pleasant time in which many distinct experiences, as coded through the Altered States of Consciousness Rating Scale (see below), were afterwards remembered, which in turn would have led to the impression of a slower passage of time. In retrospect, subjective duration expands and a slower passage of time is experienced when more vivid and changing events happened [56]. With the moderate dose of red wine, the volunteers were more absorbed and engrossed in the present moment, less aware of memories and expectations regarding the future. This again can be interpreted as an increased openness to present experience, which in retrospect would have increased the memory load and in turn would evoke the feeling of a slower passage of time. The red wine also decreased body awareness and made thoughts being perceived as passing slower, but these two effects were the smallest in statistical terms.

It is possible that the increased focus on present moment is an expression of the "myopia effect" of alcohol proposed by Steele and Josephs [57], according to whom alcohol narrows attention to salient cues, internal or external. If the salient cues are positive, increased enjoyment ensues. Others have proposed that amusing social situations can enhance mood regardless of alcohol having been drunk or not, and what alcohol does differently in social situations is to increase enjoyment in the transient moments when the attention is more internally directed and less attention is paid to the environment [58].

The red wine increased scores in the OAV subscale Insightfulness, which includes the items "I felt very profound", "I gained clarity into connections that puzzled me before", "I had very original thoughts". This is consistent with previous findings that beer and vodka can promote creativeness [23,24]. The red wine increased scores in the OAV subscale Change in Meaning of Percepts, which includes items, such as "Objects around me engaged me emotionally much more than usual", "Everyday things gained a special meaning". This appears consonant with findings that vodka enhanced the attractiveness of faces and landscapes [26], and perhaps with the capacity of alcohol to improve quickness in detecting small changes in the environment

[59]. Although alcohol can interfere with controlled attention processes, it may enhance more passive attentional processes [59] that might facilitate fascination for the present situation. Red wine also increased vividness of imagination and memories, as indicated by the OAV subscale Complex Imagery.

The red wine blurred the differentiation between the self and the environment as reflected in differences in the OAV subscale Experience of Unity. Freud [33] and many mystics have discussed the merging of the self with the external world as a common feature of mystical experience [27,30]. Items of this subscale include "Everything seemed to unify into an oneness", "It seemed to me that my environment and I were one", "I experienced a touch of eternity". In fact, the potential of red wine in moderate doses to trigger mystical-type experiences was further shown by increases in the OAV subscale "Spiritual Experience" with items such as "I had the feeling of being connected to a superior power" and "I experienced a kind of awe". This appears to confirm William James's observation that alcohol stimulates mystical faculties [37]. Rather likely, not all people will interpret sensations induced by red wine in this setting as part of the occurrence of mystical experiences with spiritual meanings, but the potential seems to be there.

It should be noted that the red wine was drunk with tranquility in a pleasant environment specialized in offering good experiences to clients. Before drinking, participants were generally in a positive mood, as can be seen in baseline pleasure. Certainly, all this contributed to the observed effects. Thus, the optimal circumstances for very positive and deep changes in consciousness elicited by red wine are given by the positive mood of drinkers, the pleasantness and complexity of the wine flavor, the pleasantness of the food that accompanies drinking, and the pleasantness of the surrounding environment of the specialized wine bar, where esthetics and entertainment play a role [3]. To the best of our knowledge, the present study was the first to explore positive changes in consciousness after a moderate dose of red wine in a naturalistic setting. Many of our predictions were confirmed, but more research is needed to corroborate our findings.

Notably, changes in consciousness were not significantly different between men and women who drank exactly the same amount of wine. We found that older people tended to report more pleasure, and younger people were more inclined to become more fascinated with the environment. It is possible that age brings greater appreciation of wine, and younger people tend to have stronger emotions in response to the esthetics of wine bars. These results must be seen as preliminary and must be replicated. Compared with Portuguese, foreigners became more immersed in the present moment. Foreigners are less familiar with the environment and with Portuguese wine; such lack of familiarity might increase the degree of immersion in novel experiences. Again, these results are preliminary and await replication. Of note, the foreigner participants were from many different countries. Future research might examine differences among particular countries and cultures that might have passed unnoticed in the comparison with the rather undifferentiated category of foreigners.

There are other directions for future studies. Unlike other alcoholic drinks, red wine is rich in polyphenols, whose ingestion can cause a greater antioxidant response than other alcoholic beverages [60]. Future research might address if the polyphenol-induced antioxidant response contributes to the specifics of alteration of consciousness caused by red wine in a fast-acting way.

Other avenues for future research consist in examining if consciousness is differently affected by different esthetical characteristics of places where people use to drink, including esthetical characteristics of different wine bars. One particular characteristic is music. Background music can affect the taste of wine [14] and increase the pleasantness of the taste experience leading to enhanced appreciation of the holistic dining experience [10,61] Additionally,

compared to a pop music background, a classical music background influenced costumers to purchase more expensive wines in a wine store [62] and to spend more in a restaurant [61]. North and colleagues explained their findings by the upmarket atmosphere of classical music inducing the congruent behavior of spending more [61]. This is certainly plausible, but another possibility is that classical music increased the pleasant experience, which in turn enhanced the appreciation of the dining experience. Future studies might examine the effect of different musical backgrounds on red wine-induced changes in consciousness.

Also, it is possible that changes in consciousness are moderated by personality traits, as well as by the esthetic elements provided by different establishments. Regarding personality, future research should focus on the role of trait absorption, which reflects individual differences in the ability to be attentionally engrossed in imagination and sensory experiences. It has been confirmed that people high on absorption are more inclined to have mystical-type experiences and to be emotionally moved by art and music [63–65]. As such, it is possible that people high on absorption are more susceptible to feel pleasant alterations of consciousness during red wine drinking experiences. There are several motives to drink alcoholic beverages, such as drinking to cope with negative mood, to enhance positive mood, to facilitate socialization, or to comply to social pressures (e.g., peer pressure) [66–68]. It would be interesting to examine how different motives influence alterations in consciousness.

The lack of a control group drinking a non-alcoholic beverage may be seen as a limitation. The advantage of the present study design in a natural setting of a bar has the disadvantage of not being able to meaningfully control for potential confounding effects, such as just enjoying sitting in the bar. The effects on the control group would have allowed us to disentangle effects of the environment per se from those of wine. It would have been interesting to assess the effect of the environment on changes in consciousness, such as in time speed, pleasure, and blissful states. Future research might also control for potential confounds, such the Hawthorne effect; participants were not observed while drinking, but a member of the research team was present in the bar to give the instructions and clarify any potential questions. As noted above, the effects of wine interacting with specific visual and auditory cues in the environment warrant future studies. It would be also interesting to examine if volunteers that come to the bar previously informed about the experiment differ from those who are contacted in the bar for the first time, that is, if some kind of a "surprise" effect does occur.

Wine drinking experiences are not detached from the environment where they occur, and in a naturalistic context, it would have been difficult to find people enjoying being in wine bar while drinking non-alcoholic beverages. Imposing that situation to participants, as in a laboratory study, would likely create an annoying artificial condition for the participants that that does fit a naturalistic design of a study in a wine bar. An experimental design with two groups of alcohol and non-alcohol drinking participants is not an ecologically valid setting in a wine bar in the heart of Lisbon, e.g., having half of the people for a period of time drinking mineral water and assessing changes in altered states of consciousness, while sitting in the bar.

The purpose of the present work was to have the wine experience in a natural setting designed to enhance pleasurableness which individuals on their own accord chose, similarly to a study on time perception under the influence of the hallucinogenic substance ayahuasca which was conducted within a shamanistic ritual [69]. The context of the *Umwelt* naturally has an effect on mood states and on the perception of time [70]. Strictly speaking, the combined effects of the setting (the bar) and the red wine cannot be dissociated. But when considering the two factors, it seems that moderate doses of red wine consumed in any pleasant surrounding will cause positive alterations in consciousness, although it is plausible that different characteristics of pleasant environments can induce different changes in consciousness, as was discussed above. Importantly, we can at least eliminate the confounder of social interaction

since all the effects were detected regardless of whether individuals drank wine alone or in company of others. A potential control condition could have been created in a laboratory setting where people drink red wine or a non-alcoholic beverage. An earlier study in a typically sterile laboratory setting showed that under the influence of alcohol, participants who drank a vodka cocktail alone felt that time was passing more quickly than participants who drank a carefully prepared control drink [71], the opposite of the present findings in a naturalistic setting. As discussed in the Introduction, increased ecological validity can be attained in many naturalistic studies where control groups are unfeasible or too detached from the real-world, such as drinking non-alcoholic drinks in a wine bar. Naturalistic studies and controlled trials should complement each other and compensate for each other limitations. Future research should aim at extending the findings of the present study with studies in other naturalistic environments and in laboratory settings. This would allow a better understanding of how environments affect the alterations of consciousness caused by red wine.

Other limitation concerns the convenience sampling of the participants including mostly university students and professionally active people below 50 years old, which does not allow us to have a representative sample of the population of wine drinkers in Portugal. Also, the fact that all volunteers had to understand English skewed the sample to a younger and more educated population. Hence, our findings might not apply to other segments of the population.

According to the present study, red wine, consumed in a moderate amount in a comfortable place, induced psychological states characterized by bliss, attentional focus on the present moment, a softening of differentiation between self and environment, fascination with the environment, original ideas, and insights, and even to some extent feelings of contact with spiritual realms. These findings contribute to a better understanding of the effects of red wine on consciousness in a cultural setting. The findings also corroborate the role of red wine as an important element of hedonism, socialization and relaxation.

## Author Contributions

**Conceptualization:** Rui Miguel Costa, Arlindo Madeira.

**Data curation:** Rui Miguel Costa.

**Formal analysis:** Rui Miguel Costa, Matilde Barata.

**Investigation:** Rui Miguel Costa, Arlindo Madeira.

**Methodology:** Rui Miguel Costa, Arlindo Madeira.

**Project administration:** Rui Miguel Costa, Arlindo Madeira.

**Resources:** Rui Miguel Costa, Arlindo Madeira.

**Supervision:** Rui Miguel Costa, Arlindo Madeira.

**Validation:** Rui Miguel Costa, Arlindo Madeira, Matilde Barata, Marc Wittmann.

**Visualization:** Rui Miguel Costa, Matilde Barata.

**Writing – original draft:** Rui Miguel Costa.

**Writing – review & editing:** Arlindo Madeira, Matilde Barata, Marc Wittmann.

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
