## [Decision Letter · Decision Letter 0]

24 Jun 2021

PONE-D-21-18138

The power of Dionysus – Effects of red wine on consciousness in a naturalistic setting

PLOS ONE

Dear Dr. Costa,

Thank you for submitting your manuscript to PLOS ONE. After careful consideration, we feel that it has merit but does not fully meet PLOS ONE’s publication criteria as it currently stands. Therefore, we invite you to submit a revised version of the manuscript that addresses the points raised during the review process.

We look forward to receiving your revised manuscript.

Kind regards,

Nikolaos Georgantzis, Dr.

Academic Editor

PLOS ONE

Journal Requirements:

1. Please ensure that your manuscript meets PLOS ONE's style requirements, including those for file naming. The PLOS ONE style templates can be found athttps://journals.plos.org/plosone/s/file?id=wjVg/PLOSOne_formatting_sample_main_body.pdf and https://journals.plos.org/plosone/s/file?id=ba62/PLOSOne_formatting_sample_title_authors_affiliations.pdf

Additional Editor Comments (if provided):

Reviewers' comments:

Reviewer's Responses to Questions

**Comments to the Author**

1. Is the manuscript technically sound, and do the data support the conclusions?

Reviewer #1: Yes

Reviewer #2: Yes

2. Has the statistical analysis been performed appropriately and rigorously? 

Reviewer #1: I Don't Know

Reviewer #2: Yes

3. Have the authors made all data underlying the findings in their manuscript fully available?

Reviewer #1: No

Reviewer #2: Yes

4. Is the manuscript presented in an intelligible fashion and written in standard English?

Reviewer #1: Yes

Reviewer #2: Yes

5. Review Comments to the Author

Reviewer #1: GENERAL OVERVIEW - The power of Dionysus – Effects of red wine on consciousness in a naturalistic setting" (PONE-D-21-18138).

This experiment in a drinking context with ecological validity permits to test theories of red wine-related effects on consciousness. The paper is very descriptive. It nevertheless represents a useful contribution to understand the effects of a moderate red wine consumption on consciousness in a naturalistic setting.

The article would benefit from a clearer definition of both the fields and the objectives behind the experiment.

Results reported are available elsewhere: https://psyarxiv.com/jc7vm/

DISCUSSION OF SPECIFIC AREAS FOR IMPROVEMENT

Sampling & recruitment

1) The use of a convenience sample lead to an over representation of certain categories of the population while others are largely under-represented. The authors should mention if their sample is representative of the drinking population in Portugal. Moreover, the fact that all participants should understand English should be mentioned as a possible limitation – as it skewed the sample to younger and more educated population (2% of unemployed, 2% of retirees, median age 33). It would be interesting to state how many potential participants have been discarded for this criterion.

2) The recruitment method should be clarified: How was the recruitment practically made? What information was given to the participants? Except from the free glasses of wine, were there other incentives for the participation in the study? It would have been interesting to distinguish the participants that came for the experiment from the ones that came to have a drink without being informed of the experiment.

4) The justification for the absence of control group is not fully satisfactory. It would have been interesting to measure the effect of the environment on elements such as Blissful state, Time speed or Pleasure.

Method

1) The selection of specific Altered States of Consciousness Rating Scale (OAV) dimensions should be mentioned and justified (elimination of: Disembodiment, Impaired control and cognition, Anxiety, Elementary imagery and Audio-visual synesthesia dimensions)

2) Considering the number of groups analyzed for variance, wouldn’t it be more appropriate to use a t-test?

3) Please include ethics statements in the methods section specifying: permits and approvals obtained for the work, including the full name of the authority that approved the study.

Data

1) A little bit of time formally setting out how the data is treated would be welcome, as some aspects remain unclear. For example, the indication of statistical significance in Table 3 could be clarified.

2) Methods and reagents are not described in sufficient detail for another researcher to reproduce the experiments described.

3) Considering that the data is self-reported consumption frequency, the reliability of this type of data should be noted, with appropriate references.

4) One of the inclusion criteria was being an adult for whom drinking two glasses of red wine was a familiar experience but more than 12% of the samples answered that they usually don’t drink red wine. This point should be clarified.

5) Considering the very small difference between the sample size and % (2 participants), is it relevant to display the frequency in the tables?

6) More details in the calculations would be appreciated

Literature

1) Some assumptions should be supported by references to the literature (e.g. connection of red wine to the appreciation of meals or to a more hedonic environment)

2) This paper could mention Steele & Josephs’ alcohol myopia theory. They posit that alcohol can lead to enjoyment by narrowing attentional capacity to stimuli in the immediate environment, and so alcohol would lead to emotional enhancement when immediate stimuli are positive (Steele & Josephs, 1990)

3) Alcohol as a promoter of creativity is not an undiscussed matter in the literature. There is a conception that a uniquely positive correlation prevails between the intake of alcohol and creativity, but only a few experimental studies address this subject. This discussion should be reflected in the literature review. See for example: Lapp, William M., et al. “On the Enhancement of Creativity by Alcohol: Pharmacology or Expectation?” The American Journal of Psychology, vol. 107, no. 2, 1994, pp. 173–206. JSTOR, www.jstor.org/stable/1423036. Accessed 21 June 2021.

As a side comment, I would recommend the authors to consider part of the literature on drinking motives and especially enhancement drinking, alcohol reward, as well as the following article:

Fairbairn, C.E., Velia, B.A., Creswell, K.G., Sayette, M.A., 2020. A dynamic analysis of the effect of alcohol consumption on humor enjoyment in a social context. Journal of Experimental Social Psychology 86, 103903. https://doi.org/10.1016/j.jesp.2019.103903

Reviewer #2: I truly appreciate Authors’ efforts to conduct a field study on the effects of red wine on insightfulness and changes in the perception of the environment. The research has several merits and is well described, allowing also non-technical readers into its development and reasoning. I only suggest Authors’ to better clarify (upfront) the limits of their final sample - not only in the discussion section - and the potential bias of the study (Hawthorne effect, confounding effects, etc.).

Major remarks

The convenience, random sample is not per se an issue however, I feel it must be clearly introduced to readers the over-representation of some categories (e.g.: students) and under-representation of other: (e.g.: retired). Additionally, the 13 individuals that stated to usually don’t drink wine do appear quite out of context in this research. A side issue is cultural background, as the final sample included tourists from several countries (and broadly 50% from Portugal) it may be that some differences among subjects are not captured with just comparing foreigners Vs. Portuguese, further studies might be suggested.

Minor remarks

To me it sounded odd the sentence (in the discussion section) “…meaningful changes in consciousness caused by a (generous) moderate”. Is it moderate or generous?

6. PLOS authors have the option to publish the peer review history of their article (what does this mean?). If published, this will include your full peer review and any attached files.

Reviewer #1: No

Reviewer #2: No

---

## [Author Response · Author response to Decision Letter 0]

22 Jul 2021

Response to reviewers

Response to Reviewer 1

Reviewer #1: GENERAL OVERVIEW - The power of Dionysus – Effects of red wine on consciousness in a naturalistic setting" (PONE-D-21-18138).

This experiment in a drinking context with ecological validity permits to test theories of red wine-related effects on consciousness. The paper is very descriptive. It nevertheless represents a useful contribution to understand the effects of a moderate red wine consumption on consciousness in a naturalistic setting.

R: Thank you.

The article would benefit from a clearer definition of both the fields and the objectives behind the experiment.

R: We provided clearer definitions of red wine and altered states of consciousness. In beginning of the first paragraph of the Introduction, we added that “Altered states of consciousness refer to substantial deviations from the habitual waking consciousness”. In the beginning of the second paragraph of the Introduction, we added that “Red wine is an alcoholic beverage made from the fermentation of dark grapes whose alcohol by volume commonly varies between 12% and 15%”.

In the last paragraph of the Introduction, we rephrased the objectives as follows: “The present study uses a naturalistic pre-post design with the objective of examining how a moderate dose of red wine induces altered states of consciousness in a group of clients of wine bars comprising mostly university students and professionally active people in their twenties, thirties, and forties. Additionally, it aims at exploring whether the changes in consciousness caused by a moderate dose of red wine differ between three conditions: drinking alone, in groups of two (dyads), and in groups between three and six persons”.

DISCUSSION OF SPECIFIC AREAS FOR IMPROVEMENT

Sampling & recruitment

1) The use of a convenience sample lead to an over representation of certain categories of the population while others are largely under-represented. The authors should mention if their sample is representative of the drinking population in Portugal. Moreover, the fact that all participants should understand English should be mentioned as a possible limitation – as it skewed the sample to younger and more educated population (2% of unemployed, 2% of retirees, median age 33). It would be interesting to state how many potential participants have been discarded for this criterion.

R: In the Discussion, we added that “Other limitation concerns the convenience sampling of the participants, which does not allow us to have a representative sample of the population of wine drinkers in Portugal. Also, the fact that all volunteers had to understand English skewed the sample to a younger and more educated population. Hence, our findings might not apply to other segments of the population”.

Additionally, we added to the Participants and procedure subsection that “The final sample size was obtained after one participant having been discarded, because of difficulties understanding the questionnaire in English”.

2) The recruitment method should be clarified: How was the recruitment practically made? What information was given to the participants? Except from the free glasses of wine, were there other incentives for the participation in the study?

R: We added to the Participants and procedure subsection that “All potential participants were approached personally by members of the research team and invited for a study investigating how a moderate amount of red wine causes changes in consciousness. They were informed that two glasses of red wine would be offered in the wine bar, and that they would just have to enjoy the moment while drinking. They were additionally informed that they would be asked to complete the questionnaire in English about changes in consciousness. Upon agreement, the participation date was scheduled. There were no other incentives for the participation in the study”.

 It would have been interesting to distinguish the participants that came for the experiment from the ones that came to have a drink without being informed of the experiment.

R: We added to the Discussion section that “It would be also interesting to examine if volunteers that come to the bar previously informed about the experiment differ from those who are contacted in the bar for the first time, that is, if some kind of a “surprise” effect does occur”.

4) The justification for the absence of control group is not fully satisfactory. It would have been interesting to measure the effect of the environment on elements such as Blissful state, Time speed or Pleasure.

R: We added to the Discussion that “It would have been interesting to assess the effect of the environment on changes in consciousness, such as in time speed, pleasure, and states of blissfulness”.

Method

1) The selection of specific Altered States of Consciousness Rating Scale (OAV) dimensions should be mentioned and justified (elimination of: Disembodiment, Impaired control and cognition, Anxiety, Elementary imagery and Audio-visual synesthesia dimensions)

R: We added to the Measures subsection that “For the purpose of the present study we excluded other dimensions of the Altered States of Consciousness Rating Scale (OAV): Disembodiment, Anxiety, Impaired Control and Cognition, Elementary Imagery, Audio-Visual Synesthesias. The reason is that we did not want to burden participants with questions about unlikely changes such as panic or disembodiment experiences, and because we did not expect that many participants sitting in the wine bar, often socializing, would be attentive to alterations in mental imagery by staying a long time with closed the eyes or otherwise in a mental state very detached from the social environment”.

2) Considering the number of groups analyzed for variance, wouldn’t it be more appropriate to use a t-test?

R: In Statistical analyses subsection, we added a reference (Altman & Bland, 1996) to support that ANOVAs are more appropriate when three or more groups are compared.

3) Please include ethics statements in the methods section specifying: permits and approvals obtained for the work, including the full name of the authority that approved the study.

R: We added to the Methods section (Participants and procedure subsection) that “The study received the approval of the Ethics Committee of ISPA – Instituto Universitário”.

Data

1) A little bit of time formally setting out how the data is treated would be welcome, as some aspects remain unclear. For example, the indication of statistical significance in Table 3 could be clarified.

R: We now indicate the statistical significance with asterisks. Nonsignificant relationships are indicated by "NS".

2) Methods and reagents are not described in sufficient detail for another researcher to reproduce the experiments described.

R: It is unclear what the “reagents” mean in the present context. Nevertheless, we added detailed information of the process of wine choice and features of the environment. Regarding the choice of wine, we added to the Participants and procedure subsection that “The choice of this particular wine was made by the producer and by the two sommeliers of the bar, where the experiment was conducted, based on their know-how of the market. It is a consensual wine with a new world profile and origin in a hot terroir, where grapes have greater maturation. This allows silky, full-bodied wines with a lesser degree of acidity, which are enjoyed by the majority of people. During the two weeks preceding the study, several clients of the bar were offered this wine (for free) for a blind tasting, and the feedback was positive and unanimous relative to its quality”. Regarding the features of the environment, we added to the Participants and Procedure subsection that “The bar has an intimate atmosphere with clients talking in low voice. There is a maximum occupation of 28 people shared by six tables, each one allowing between four and six people. There are no standing clients”.

3) Considering that the data is self-reported consumption frequency, the reliability of this type of data should be noted, with appropriate references.

R: We now note that “Descriptive statistics are displayed in Table 1, including the self-reported frequency of alcohol and red wine consumption, which are considered reliable measures”. This was supported by references 39 (Williams et al., 1985) and 40 (Gonbaek & Heitmann, 1996).

4) One of the inclusion criteria was being an adult for whom drinking two glasses of red wine was a familiar experience but more than 12% of the samples answered that they usually don’t drink red wine. This point should be clarified.

R: We added to the Participants and procedure subsection that “Thirteen participants reported that they usually do not drink red wine (see Table 1), but before the experiment we confirmed that for them red wine was occasionally drank and was a familiar experience”. 

5) Considering the very small difference between the sample size and % (2 participants), is it relevant to display the frequency in the tables?

R: We deleted the frequency in the tables.

6) More details in the calculations would be appreciated

R: The statistical analyses provide the necessary information for replication and interpretation, including effect sizes. As stated above, the statistical procedure is now backed by literature, as suggested.

Literature

1) Some assumptions should be supported by references to the literature (e.g. connection of red wine to the appreciation of meals or to a more hedonic environment)

R: We added references 4 (Fiore et al., 2020), 5 (Galmarini 2020), 6 (Olsen et al., 2007) and 7 (Yu et al., 2009).

2) This paper could mention Steele & Josephs’ alcohol myopia theory. They posit that alcohol can lead to enjoyment by narrowing attentional capacity to stimuli in the immediate environment, and so alcohol would lead to emotional enhancement when immediate stimuli are positive (Steele & Josephs, 1990)

R: We added to the Discussion that “It is possible that the increased focus on present moment is an expression of the “myopia effect” of alcohol proposed by Steele and Josephs [57], according to whom alcohol narrows attention to salient cues, internal or external. If the salient cues are positive, increased enjoyment ensues”.

3) Alcohol as a promoter of creativity is not an undiscussed matter in the literature. There is a conception that a uniquely positive correlation prevails between the intake of alcohol and creativity, but only a few experimental studies address this subject. This discussion should be reflected in the literature review. See for example: Lapp, William M., et al. “On the Enhancement of Creativity by Alcohol: Pharmacology or Expectation?” The American Journal of Psychology, vol. 107, no. 2, 1994, pp. 173–206. JSTOR, www.jstor.org/stable/1423036. Accessed 21 June 2021.

R: In the Introduction, we added that “alcohol-related creativity was also found to be enhanced by expectations rather than actual effects of alcohol”, and cited the study by Lapp and colleagues.

As a side comment, I would recommend the authors to consider part of the literature on drinking motives and especially enhancement drinking, alcohol reward, as well as the following article:

Fairbairn, C.E., Velia, B.A., Creswell, K.G., Sayette, M.A., 2020. A dynamic analysis of the effect of alcohol consumption on humor enjoyment in a social context. Journal of Experimental Social Psychology 86, 103903. https://doi.org/10.1016/j.jesp.2019.103903

R: We added to the Discussion that “There are several motives to drink alcoholic beverages, such as drinking to cope with negative mood, to enhance positive mood, to facilitate socialization, or to comply to social pressures (e.g., peer pressure). It would be interesting to examine how different motives influence alterations in consciousness”. In this regard, we added references 66 (Littlefied et al., 2010), 67 (Kunstsche et al., 2005) and 68 (Bresin & Mekawi, 2021). 

We added to the Discussion that “Others have proposed that amusing social situations can enhance mood regardless of alcohol having been drunk or not, and what alcohol does differently in social situations is to increase enjoyment in the transient moments when the attention is more internally directed and less attention is paid to the environment”. We support this sentence with the reference to Fairbairn and colleagues.

We added to the Discussion that “Such activation of the dopaminergic neurons is consistent with the documented effects of a moderate dose of alcohol on the reward system by which incentive salience is increased”. In this regard, we added references 53 (Ingvar et al., 1998) and 54 (Charlet et al., 2013).

Response to Reviewer 2

Reviewer #2: I truly appreciate Authors’ efforts to conduct a field study on the effects of red wine on insightfulness and changes in the perception of the environment. The research has several merits and is well described, allowing also non-technical readers into its development and reasoning.

R: Thank you.

I only suggest Authors’ to better clarify (upfront) the limits of their final sample - not only in the discussion section - and the potential bias of the study (Hawthorne effect, confounding effects, etc.).

Major remarks

The convenience, random sample is not per se an issue however, I feel it must be clearly introduced to readers the over-representation of some categories (e.g.: students) and under-representation of other: (e.g.: retired). 

R: In the last paragraph of the Introduction, we now state that “The present study uses a naturalistic pre-post design with the objective of examining how a moderate dose of red wine induces altered states of consciousness in a group of clients of wine bars comprising mostly university students and professionally active people in their twenties, thirties, and forties”.

In the Discussion, we added that “Other limitation concerns the convenience sampling of the participants including mostly university students and professionally active people below 50 years old, which does not allow us to have a representative sample of the population of wine drinkers in Portugal. Also, the fact that all volunteers had to understand English skewed the sample to a younger and more educated population. Hence, our findings might not apply to other segments of the population”. 

We also added to the Discussion that “It would have been interesting to assess the effect of the environment on changes in consciousness, such as in time speed, pleasure, and blissful states. Future research might also control for potential confounds, such the Hawthorne effect; participants were not observed while drinking, but a member of the research team was present in the bar to give the instructions and clarify any potential questions. As noted above, the effects of wine interacting with specific visual and auditory cues in the environment warrants future studies. It would be also interesting to examine if volunteers that come to the bar previously informed about the experiment differ from those who are contacted in the bar for the first time, that its, if some kind of a “surprise” effect does occur”.

Additionally, the 13 individuals that stated to usually don’t drink wine do appear quite out of context in this research. 

R: In addressing a similar concern of the Reviewer 1, we stated that “Thirteen participants reported that they usually do not drink red wine (see Table 1), but before the experiment we confirmed that for them red wine was a familiar experience, because it is occasionally drank”.

A side issue is cultural background, as the final sample included tourists from several countries (and broadly 50% from Portugal) it may be that some differences among subjects are not captured with just comparing foreigners Vs. Portuguese, further studies might be suggested.

R: We added to the Discussion that “Of note, the foreigner participants were from many different countries. Future research might examine differences among particular countries and cultures that might have passed unnoticed in the comparison with the rather undifferentiated category of foreigners”.

Minor remarks

To me it sounded odd the sentence (in the discussion section) “…meaningful changes in consciousness caused by a (generous) moderate”. Is it moderate or generous?

R: We deleted “generous”.

---

## [Decision Letter · Decision Letter 1]

2 Aug 2021

The power of Dionysus – Effects of red wine on consciousness in a naturalistic setting

PONE-D-21-18138R1

Dear Dr. Costa,

We’re pleased to inform you that your manuscript has been judged scientifically suitable for publication and will be formally accepted for publication once it meets all outstanding technical requirements.

Kind regards,

Nikolaos Georgantzis, Dr.

Academic Editor

PLOS ONE

Additional Editor Comments (optional):

Reviewers' comments:

Reviewer's Responses to Questions

**Comments to the Author**

1. If the authors have adequately addressed your comments raised in a previous round of review and you feel that this manuscript is now acceptable for publication, you may indicate that here to bypass the “Comments to the Author” section, enter your conflict of interest statement in the “Confidential to Editor” section, and submit your "Accept" recommendation.

Reviewer #1: All comments have been addressed

Reviewer #2: All comments have been addressed

2. Is the manuscript technically sound, and do the data support the conclusions?

Reviewer #1: Yes

Reviewer #2: Yes

3. Has the statistical analysis been performed appropriately and rigorously? 

Reviewer #1: Yes

Reviewer #2: Yes

4. Have the authors made all data underlying the findings in their manuscript fully available?

Reviewer #1: Yes

Reviewer #2: Yes

5. Is the manuscript presented in an intelligible fashion and written in standard English?

Reviewer #1: Yes

Reviewer #2: Yes

6. Review Comments to the Author

Reviewer #1: (No Response)

Reviewer #2: All issues have been succesfully addressed by scholars in this round of revisions. Therefore, I suggest pubblication of the paper

7. PLOS authors have the option to publish the peer review history of their article (what does this mean?). If published, this will include your full peer review and any attached files.

Reviewer #1: No

Reviewer #2: No

---

## [Editor Report · Acceptance letter]

13 Aug 2021

PONE-D-21-18138R1 

The power of Dionysus – Effects of red wine on consciousness in a naturalistic setting 

Dear Dr. Costa:

I'm pleased to inform you that your manuscript has been deemed suitable for publication in PLOS ONE. Congratulations! Your manuscript is now with our production department. 

Kind regards, 

on behalf of

Prof. Nikolaos Georgantzis 

Academic Editor

PLOS ONE